# Scalable Differentially Private Data Generator via Private Aggregation of Teacher Discriminators

## Abstract

We present a novel approach G-PATE for training a scalable differentially private data generator, which can be used to produce synthetic datasets with strong privacy guarantee while preserving high data utility. Our approach leverages generative adversarial nets to generate data and exploits the PATE (Private Aggregation of Teacher Ensembles) framework to protect data privacy. Compared to existing methods, our approach significantly improves the use of privacy budget. This is possible since we only need to ensure differential privacy for the generator, which is the part of the model that actually needs to be published for private data generation. In particular, we connect a student generator with an ensemble of teacher discriminators and propose a private gradient aggregation mechanism to ensure differential privacy on all the information that flows from the teacher discriminators to the student generator. Theoretically, we prove that our algorithm ensures differential privacy for the generator. Empirically, we provide thorough experiments to demonstrate the superiority of our method over prior work on both image and non-image datasets.

## 1 Introduction

Machine learning has been applied to a wide range of applications such as face recognition (Parkhi et al., 2015), autonomous driving (Menze & Geiger, 2015), and medical diagnoses (de Bruijne, 2016; Kourou et al., 2015). However, most learning methods rely on the availability of large-scale training datasets containing sensitive information such as personal photos or medical records. Therefore, such sensitive datasets are often hard to be shared due to privacy concerns. To handle this challenge, data providers sometimes release synthetic datasets produced by generative models learned on the original data. Though recent studies show that generative models such as generative adversarial networks (GAN) (Goodfellow et al., 2014) can generate synthetic records that are indistinguishable from the original data distribution, there is no theoretical guarantee on the privacy protection. While privacy definitions such as differential privacy (Dwork & Feldman, 2018) and Rényi differential privacy (Mironov, 2017) provide rigorous privacy guarantee, applying them to synthetic data generation is nontrivial.

Recently, two approaches have been proposed to combine differential privacy with synthetic data generation: DP-GAN (Xie et al., 2018) and PATE-GAN (Yoon et al., 2019). DP-GAN modifies GAN by training the discriminator using differentially private stochastic gradient descent. Though it achieves privacy guarantee due to the post processing property (Dwork et al., 2014) of differential privacy, DP-GAN incurs significant utility loss on the synthetic data, especially when the privacy budget is low. In contrast, PATE-GAN trains differentially private GAN using the PATE mechanism (Papernot et al., 2017). Specifically, it first trains a set of teacher discriminators and then train a student discriminator based on the trained ensemble of teacher discriminators. To ensure differential privacy, the student discriminator is only trained on records that are produced by the generator and labeled by the teacher discriminators. The key limitation of this approach is that it relies on the assumption that the generator would be able to generate the entire "real" records space to bootstrap the training process. If most of the synthetic records are labeled as fake by the teacher discriminators, the student discriminator would be trained on a biased dataset and fail to learn the true data distribution. Consequently, this trained generator would not be able to produce high-quality synthetic data. This problem does not exist on a traditional GAN, where the discriminator is always

able to provide useful information to the generator since they can access the *real data records* rather than the synthetic data only. In addition, the two stage training process of PATE-GAN makes it less scalable or flexible in terms of varying the number of teacher discriminators.

The main contribution of this paper is a new approach named G-PATE for training a differentially private data generator by combining the generative model with PATE mechanism. Our approach is based on the key observation that: *It is not necessary to ensure differential privacy for the discriminator in order to train a differentially private generator.* As long as we ensure differential privacy on the information flow from the discriminator to the generator, it is sufficient to guarantee the privacy property for the generator. To achieve this, we propose a private gradient aggregation mechanism to ensure differential privacy on all the information that flows from the teacher discriminators to the student generator. Compared to PATE-GAN, our approach has three advantages. First, we improve the use of privacy budget by only applying it to the part of the model that actually needs to be released for data generation. Second, our discriminator can be trained on original data records since it does not need to satisfy differential privacy. Finally, G-PATE is much more scalable given its simple framework architecture.

Theoretically, we show that our algorithm ensures differential privacy for the generator. Empirically, we conduct extensive experiments on the standard Kaggle credit card fraud detection dataset, as well as two image datasets MNIST and Fashion-MNIST. The results show that our method significantly outperforms all baselines including DP-GAN and PATE-GAN in terms of data utility.

## 2 RELATED WORK

Differential privacy (Dwork, 2008) is a notion that ensures an algorithm outputs general information about its input dataset without revealing individual information. Rényi differential privacy (Mironov, 2017) is a relaxation of differential privacy that allows tighter analysis on the composition of privacy budgets. Following the work of differential privacy, researchers have proposed different methods to design differentially private statistical functions and machine learning models (Bassily et al., 2014; Chaudhuri et al., 2011; Abadi et al., 2016; McSherry & Talwar, 2007; Friedman & Schuster, 2010). Recently, various approaches have been proposed for differentially private data generation. Priview (Qardaji et al., 2014) generates synthetic data based on marginal distributions of the original dataset, and PrivBayes (Zhang et al., 2017) trains a differentially private Bayesian network. However, these approaches are not suitable for image datasets since the statistics they use cannot well preserve the correlations between pixels in an image.

Both DP-GAN (Xie et al., 2018) and PATE-GAN (Yoon et al., 2019) apply differential privacy to the training process of generative adversarial networks. They both ensure differential privacy while training the discriminator, and the privacy property of the generator is guaranteed by the post processing property of differential privacy. Unlike their approaches, our method improves the use of privacy budget by only ensuring differential privacy on the generator, which is the part that actually needs to be released for synthetic data generation. This improvement allows us to incur much lower utility loss on the synthetic data under the same privacy constraint.

Private aggregation of teacher ensembles (PATE) is a method to train a differentially private classifier using ensemble mechanisms. It first trains an ensemble of teacher models on disjoint subsets of the sensitive training data. Then, a differentially private student model is trained on public data labeled by the teacher models. The privacy guarantee of PATE is more intuitive to understand because no teacher model can dictate the training of the student model. Compared to most differentially private classification method, PATE benefits from a tighter data-dependent privacy bound especially when teacher models are likely to reach consensus. Scalable PATE (Papernot et al., 2018) improves the utility of PATE with a Confident-GNMax aggregator that only returns a result if it has high confidence in the consensus among teachers. However, both PATE and Scalable PATE relies on the availability of public unlabeled data, and their aggregators are only applicable to categorical data (i.e., class labels). In contrast, our method does *not* rely on any public dataset and can generate synthetic samples that are differentailly private with respect to the private training dataset. We also design a differentially private gradient aggregator that works for continuous gradient vectors.

## 3 BACKGROUND

We recall the definition of differential privacy (DP), Rényi differential privacy (RDP), and some of their properties.

### 3.1 DIFFERENTIAL PRIVACY

Differential privacy bounds the shift in the output distribution of a randomized algorithm that could be caused by a small input perturbation. The following definition formally describes this privacy guarantee.

**Definition 1** (($\varepsilon, \delta$)-Differential Privacy). A randomized algorithm $\mathcal{M}$ with domain $\mathbb{N}^{|\mathcal{X}|}$ is ($\varepsilon, \delta$)-differentially private if for all $\mathcal{S} \subseteq \mathrm{Range}(\mathcal{M})$ and for any neighboring datasets $D$ and $D'$:

$$\Pr[\mathcal{M}(D) \in \mathcal{S}] \leq \exp(\varepsilon) \Pr[\mathcal{M}(\mathcal{D}') \in \mathcal{S}] + \delta.$$

### 3.2 RÉNYI DIFFERENTIAL PRIVACY

Rényi differential privacy is a natural relaxation of differential privacy. Defined below, its privacy guarantee is expressed in terms of Rényi divergence.

**Definition 2** (($\lambda, \varepsilon$)-RDP). A randomized mechanism $\mathcal{M}$ is said to guarantee ($\lambda, \varepsilon$)-RDP with $\lambda > 1$ if for any neighboring datasets $D$ and $D'$,

$$D_\lambda \left( \mathcal{M}(D) \| \mathcal{M}(D') \right) = \frac{1}{\lambda - 1} \log \mathbb{E}_{x \sim \mathcal{M}(D)} \left[ \left( \frac{\mathbf{Pr}[\mathcal{M}(D) = x]}{\mathbf{Pr}[\mathcal{M}(D') = x]} \right)^{\lambda - 1} \right] \leq \varepsilon.$$

($\lambda, \varepsilon$)-RDP implies ($\varepsilon_\delta, \delta$)-differential privacy for any given probability $\delta > 0$.

**Theorem 1** (From RDP to DP). *If a mechanism $\mathcal{M}$ guarantees ($\lambda, \varepsilon$)-RDP, then $\mathcal{M}$ guarantees $(\varepsilon + \frac{\log 1/\delta}{\lambda - 1}, \delta)$-differential privacy for any $\delta \in (0, 1)$.*

Compared to DP, RDP supports easier composition of multiple queries and clearer privacy guarantee under Gaussian noise. Specifically, RDP could be easily composed by adding the privacy budget:

**Theorem 2** (Composition of RDP). *If a mechanism $\mathcal{M}$ consists of a sequence of $\mathcal{M}_1, \ldots, \mathcal{M})k$ such that for any $i \in [k]$, $\mathcal{M}_i$ guarantees ($\lambda, \varepsilon_i$)-RDP, then $\mathcal{M}$ guarantees ($\lambda, \sum_{i=1}^k \varepsilon_i$)-RDP.*

Suppose $f$ is a real-valued function, and the Gaussian mechanism is defined as follows:

$$\mathbf{G}_\sigma f(D) = f(D) + N \left( 0, \sigma^2 \right),$$

where $N \left( 0, \sigma^2 \right)$ is normally distributed random variable with standard deviation $\sigma$ and mean 0. The Gaussian mechanism provides the following RDP guarantee:

**Theorem 3** (RDP Guarantee for Gaussian Mechanism). *If $f$ has sensitivity 1, then the Gaussian mechanism $\mathbf{G}_\sigma f$ satisfies $\left( \lambda, \lambda / \left( 2\sigma^2 \right) \right)$-RDP.*

## 4 THE G-PATE METHOD

In this section, we present our method named G-PATE. An overview of the method is shown in Figure 1. Unlike PATE-GAN and DP-GAN, G-PATE ensures differential privacy for the information flow from the discriminator to the generator. This improvement incurs less utility loss on the synthetic samples, so it can generate synthetic samples for higher dimensional and more complex datasets.

G-PATE makes two major modifications on the training process of GAN. First, we replace the discriminator in GAN with an ensemble of teacher discriminators trained on disjoint subsets of the sensitive data. The teacher discriminators do not need to be published, thus can be trained using non-private algorithms. In addition, we design a gradient aggregator to collect information from teacher discriminators and combine them in a differentially private fashion. The output of the aggregator is a gradient vector that guides the student generator to improve its synthetic samples.

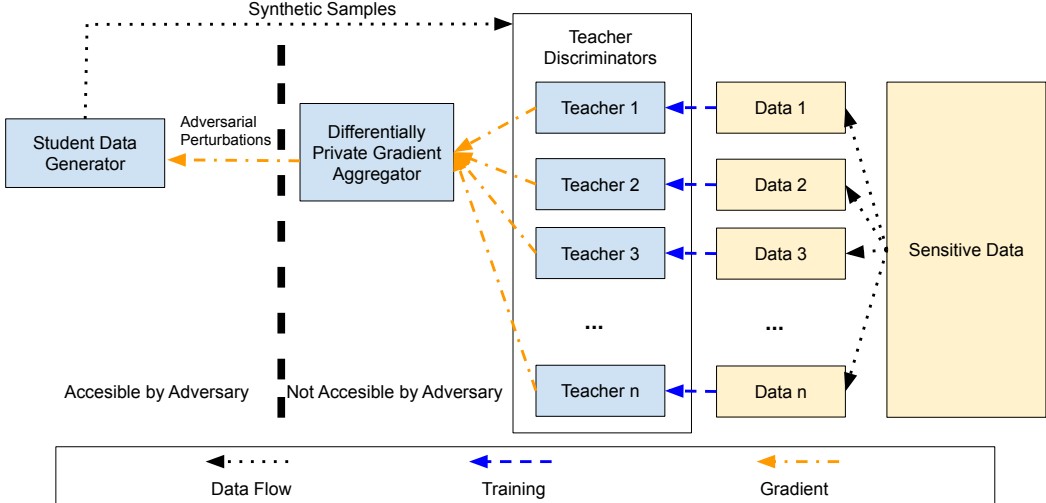

Figure 1: **Model Overview of G-PATE.** The model contains three parts: a student data generator, a differentially private gradient aggregator, and an ensemble of teacher discriminators.

Unlike PATE-GAN, G-PATE does not require any student discriminator. The teacher discriminators are directly connected to the student generator. The gradient aggregator sanitizes the information flow from the teacher discriminators to the student generator to ensure differential privacy. This way, G-PATE uses privacy budget more efficiently and better approximates the real data distribution to ensure high data utility.

### 4.1 TRAINING THE STUDENT GENERATOR

To achieve better privacy budget efficiency, G-PATE only ensures differential privacy for the generator and allows the discriminators to learn private information. The privacy property is achieved by sanitizing all information propagated from the discriminators to the generator. To ease privacy analysis, we decompose G-PATE into three parts: the teacher discriminators, the student generator, and the gradient aggregator. To prevent the propagation of private information, the student generator does not have direct access to any information in any of the teacher discriminators. Consequently, we cannot train the student generator by ascending its gradient based on loss of the discriminators. To solve this problem, we propose the use of *adversarial perturbation*, which is a small manipulation on the fake record $x$ that causes the discriminator's loss on $x$ to increase. The adversarial perturbation is calculated by ascending $x$'s gradients on the loss of the discriminator. It teaches the student generator how to improve its fake records. In each training iteration, the student generator is updated in three steps: (1) A teacher discriminator generates adversarial perturbations for each record produced by the student generator. (2) The gradient aggregator takes the adversarial perturbations from all teacher models and generates a differentially private aggregation of them. (3) The student generator updates its weights based on the privately aggregated adversarial perturbation. The process is formally presented in Algorithm 1.

**Generating Adversarial Perturbations.** Let $D$ be a teacher discriminator. Given a fake record $x$, we use $\mathcal{L}_D(x)$ to represent $D$'s loss on $x$. In each training iteration, the weights of $D$ are updated by descending their stochastic gradients on $\mathcal{L}_D$.

For each input fake record $x$, we generate an adversarial perturbation $\Delta x$ that guides the student generator on improving its output. By applying the perturbation on its output, the student generator would get an improved fake record $\hat{x} = x + \Delta x$ on which $D$ has a higher loss. Therefore, $\Delta x$ is calculated as $x$'s gradients on $\mathcal{L}_D$:

$$\Delta x = \left. \frac{\partial \mathcal{L}_D(a)}{\partial a} \right|_{a=x}. \tag{1}$$

---

**Algorithm 1 - Training the Student Generator.** The student generator is jointly trained with an ensemble of teacher discriminators. In each iteration, the student generator updates its weights based on an aggregated adversarial perturbation generated by the teacher ensemble.

---

**Require:** batch size $m$, number of teacher models $n$, number of training iterations $N$, gradient clipping constant $c$, number of bins $B$, projected dimension $k$, noise parameters $\sigma_1$ and $\sigma_2$, threshold $T$, disjoint subsets of sensitive data $d_1, d_2, \ldots, d_n$
 1: **for** number of training iterations **do**
 2:     Sample $m$ noise samples $\{z_1, z_2, \ldots, z_m\}$
 3:     Generate fake samples $\{G(z_1), G(z_2), \ldots, G(z_m)\}$
 4:     **for** $i \in \{1, \ldots, n\}$ **do**
 5:         Sample $m$ data samples $\{x_1, x_2, \ldots, x_m\}$ from $d_i$
 6:         Update the teacher discriminator $D_i$ by descending its stochastic gradient on $\mathcal{L}_{D_i}$ on both fake samples and real samples
 7:         **for** $j \in \{1, \ldots, m\}$ **do**
 8:             Calculate the adversarial perturbation $\Delta x_j^{(i)}$ as $x_j$'s gradients on $\mathcal{L}_{D_i}(x_j)$
 9:         **end for**
10:     **end for**
11:     **for** $j \in \{1, \ldots, m\}$ **do**
12:         $\Delta x_j \leftarrow \texttt{DPGradAgg}(\Delta x_j^{(1)}, \Delta x_j^{(2)}, \ldots, \Delta x_j^{(n)}, c, B, k, \sigma_1, \sigma_2, T)$
13:         $\hat{x}_j \leftarrow G(z_j) + \Delta x_j$
14:     **end for**
15:     Update the student generator $G$ by descending its stochastic gradient on $\mathcal{L}_G$ on $\{\hat{x}_1, \hat{x}_2, \ldots, \hat{x}_m\}$
16: **end for**
17: **return** G

---

With the adversarial perturbation $\Delta x$, the student generator can be trained without direct access to the discriminator's loss.

**Updating the Student Generator.** A student generator $G$ learns to map a random input $z$ to a fake record $x = G(z)$ so that $x$ is indistinguishable from a real record by $D$. Given an adversarial perturbation $\Delta x$, the teacher discriminators have higher loss on the perturbed fake record $\hat{x} = x + \Delta x$ compared to the original fake record $x$. Therefore, the student generator learns to improve its fake records by minimizing the mean squared error (MSE) between its output $G(z)$ and the perturbed fake record $\hat{x}$.

$$\mathcal{L}_G(z, \hat{x}) = \frac{1}{k} \sum_{i=1}^{k} (G(z)_i - \hat{x}_i)^2, \tag{2}$$

where $k$ is the number of synthetic records generated per training iteration. To ensure differential privacy, instead of receiving the adversarial perturbation from a single discriminator, we train the student generator using a differentially private gradient aggregator that combines adversarial perturbations from multiple teacher discriminators. Details are provided in Section 4.2.

## 4.2 DIFFERENTIALLY PRIVATE GRADIENT AGGREGATION FOR G-PATE

G-PATE consists of a student generator and an ensemble of teacher discriminators trained on disjoint subsets of the sensitive data. In each training iteration, each teacher discriminator generates an adversarial perturbation $\Delta x$ that guides the student generator on improving its output records. Different from traditional GAN, in G-PATE, the student generator does not have access to the loss of any teacher discriminators, and the adversarial perturbation is the only information propagated from the teacher discriminators to the student generator. Therefore, to achieve differential privacy, it suffices to add noise during the aggregation of the adversarial perturbations.

However, the aggregators used in PATE and PATE-GAN are not suitable for aggregating gradient vectors because they are only applicable to categorical data. Therefore, we propose a differentially private gradient aggregator (`DPGradAgg`) based on PATE. With gradient discretization, we convert gradient aggregation into a voting problem and get the noisy aggregation of teachers' votes using

PATE. Additionally, we use random projection to reduce the dimension of vectors on which the aggregation is performed. The combination of these two approaches allows G-PATE to generate synthetic samples with higher data utility, even for large scale image datasets, which is hard to be achieved by PATE-GAN. The procedure is formally presented in Algorithm 2.

---

**Algorithm 2 - Differentially Private Gradient Aggregator (`DPGradAgg`).** This algorithm takes a list of gradient vectors and returns a differentially private aggregation of them.

---

**Require:** gradient vectors $\{\Delta x^{(1)}, \Delta x^{(2)}, \dots, \Delta x^{(n)}\}$, gradient clipping constant $c$, number of bins $B$, projected dimension $k$, noise parameters $\sigma_1$ and $\sigma_2$, threshold $T$
1: $k_0 \leftarrow$ the dimension of $\Delta x^{(1)}$
2: $R \leftarrow$ a $k_0$ by $k$ random projection matrix with each component randomly drawn from $\mathcal{N}(0, \frac{1}{k})$
3: $\{\Delta u^{(1)}, \Delta u^{(2)}, \dots, \Delta u^{(n)}\} \leftarrow \{\Delta x^{(1)}R, \Delta x^{(2)}R, \dots, \Delta x^{(n)}R\}$
4: $\Delta u \leftarrow$ empty list
5: **for** $j \in 1, 2, \dots, k$ **do**
6: $\quad v \leftarrow$ a vector containing the $j$th element of all gradients in $\{\Delta u^{(1)}, \Delta u^{(2)}, \dots, \Delta u^{(n)}\}$
7: $\quad$ Clip $v$ to $(-c, c)$
8: $\quad h \leftarrow$ the histogram of $v$ with $B$ bins of width $\frac{2c}{B}$
9: $\quad j \leftarrow$ `Confident-GNMax`$(h, \sigma_1, \sigma_2, T)$
10: $\quad$ Append the midpoint of the $j$-th bin to $\Delta u$
11: **end for**
12: $\Delta x \leftarrow \Delta u R^T$
13: **return** $\Delta x$

---

**Gradient Discretization.** Since PATE is originally designed for aggregating the teacher models' votes on the correct class label of an example, the aggregation mechanism in PATE only applies to categorical data. Therefore, we design a three-step algorithm to apply PATE on continuous gradient vectors. First, we discretize the gradient vector by creating a histogram and mapping each element to the midpoint of the bin it belongs to. Then, instead of voting for the class labels as in PATE, a teacher discriminator votes for $k$ bins associated with $k$ elements in its gradient vector. Finally, for each dimension, we calculate the bin with most votes using the `Confident-GNMax` aggregator (Papernot et al., 2018) (Algorithm 3 in Appendix A.4). The aggregated gradient vector is consisted of the midpoints of the selected bins.

With gradient discretization, the teacher discriminators can directly communicate with the student generator using the PATE mechanism. Since these teacher discriminators are trained on real data, they can provide much better guidance to the generator compared to the student discriminator in PATE-GAN, which is only trained on synthetic samples. Moreover, the `Confident-GNMax` aggregator ensures that the student generator would only improve its output in the direction agreed by most of the teacher discriminators.

**Random Projection.** Aggregation of high dimensional vectors is expensive in terms of privacy budget because private voting needs to be performed on each dimension of the vectors. To save privacy budget, we use random projection (Bingham & Mannila, 2001) to reduce the dimensionality of gradient vectors. Before the aggregation, we generate a random projection matrix with each component randomly drawn from a Gaussian distribution. We then project the gradient vector into a lower dimensional space using the random projection matrix. After the aggregation, the aggregated gradient vector is projected back to its original dimensions. Since the generation of random projection matrix is data-independent. It does not consume any privacy budgets.

Random projection is shown to be especially effective on image datasets. Since different pixels of an image are often highly correlated, the intrinsic dimension of an image is usually much lower than the number of pixels (Gong et al., 2018). Therefore, random projection maximizes the amount of information a student generator can get from a single query to the `Confident-GNMax` aggregator, and makes it possible for G-PATE to retain reasonable utility even on high dimensional data. Moreover, random projection preserves similar squared Euclidean distance between high-dimensional vectors, therefore is beneficial to privacy protection both theoretically and empirically (Xu et al., 2017).

## 5 THEORETICAL GUARANTEES

In this section, we provide theoretical guarantees on the privacy properties for G-PATE. To start with, we propose the following definition for a differentially private data generative model.

**Definition 3** (Differentially Private Generative Model). Let $G$ be a generative model that maps a set of points $Z$ in the noise space $\mathcal{Z}$ to a set of records $X$ in the data space $\mathcal{X}$. Let $\mathcal{D}$ be the training dataset of $G$ and $\mathcal{A} : \mathcal{D} \mapsto G$ be the training algorithm. We say that $G$ is a $(\varepsilon, \delta)$-*differentially private data generative model* if the training algorithm $\mathcal{A}$ is $(\varepsilon, \delta)$-differentially private.

Definition 3 relaxes the definition of a DP-GAN by focusing the protection only on the generative model in a GAN. This relaxation saves privacy budget during training and improves the utility of the model. Moreover, the relaxation does not compromise the privacy guarantee for the synthetic data.

Theorem 4 shows that a differentially private generative model is able to support infinite number of queries to data generator and can be used to generate multiple synthetic datasets.

**Theorem 4.** *Let $G$ be an $(\varepsilon, \delta)$-differentially private data generative model trained on a private dataset $\mathcal{D}$. For any $Z \in \mathcal{Z}$, the synthetic dataset $X = G(Z)$ is $(\varepsilon, \delta)$-differentially private.*

Theorem 4 is a consequence of the post-processing property of differential privacy. First, the random points $Z$ are independent of the private dataset $\mathcal{D}$. Second, one does not need to query the discriminator during the data generation process. Therefore, the synthetic dataset is generated by post-processing $G$ and is guaranteed to be $(\varepsilon, \delta)$-differentially private.

The next theorem justifies the Rényi differential privacy of the G-PATE method.

**Theorem 5** (Rényi Differential Privacy of G-PATE). *Let $\mathcal{A}$ be the training algorithm for the student generator (Algorithm 1) with $N$ training iterations and $k$ projected dimensions. The data-dependent Rényi differential privacy for $\mathcal{A}$ with order $\lambda > 1$ is $\varepsilon = \sum_{1 \leq i \leq N} \left( \sum_{1 \leq j \leq k} \varepsilon_{i,j} \right)$, where $\varepsilon_{i,j}$ is the data-dependent Rényi differential privacy for the* `Confident-GNMax` *aggregator in the $i$-th iteration on the $j$-th projected dimension.*

Theorem 5 holds because of the composition theorem of Rényi differential privacy (Theorem 2 in Appendix A.4). During the training process, the student generator accesses information about the sensitive dataset through the `Confident-GNMax` aggregator. It submits $k$ queries per training iteration. Therefore, the Rényi differential privacy budget of the training algorithm is a composition of the data-dependent Rényi differential privacy budget of the `Confident-GNMax` aggregator over $k$ dimensions and $N$ iterations. The data-dependent privacy budget for each `Confident-GNMax` aggregation is dependent on $\sigma_1$, $\sigma_2$, and threshold $T$. We include the analysis in Appendix A.4. The remaining parameters (e.g. gradient clipping constant $c$, number of bins $B$) do not influence the privacy guarantee.

The next theorem provides a theoretical guarantee on the differential privacy of the G-PATE method.

**Theorem 6** (Differential Privacy of G-PATE). *Given a sensitive dataset $\mathcal{D}$ and parameters $0 < \delta < 1$, let $G$ be the student generator trained by Algorithm 1. There exists $\varepsilon > 0$ and $\lambda > 1$ so that $G$ is a $(\varepsilon + \frac{\log 1/\delta}{\lambda - 1}, \delta)$-differentially private data generative model.*

Theorem 6 is the consequence of converting the Rényi differential privacy guarantee in Theorem 5 to differential privacy (Theorem 1 in Appendix A.4).

## 6 EXPERIMENTAL EVALUATION

We evaluate G-PATE against two state-of-the-art benchmarks: DP-GAN and PATE-GAN. To compare the performance of different data generators, we train a classifier on the synthetic data and test it on the original data to benchmark the quality of the synthetic data.

We first perform comparative analysis with PATE-GAN and DP-GAN on the datasets used in the corresponding works (i.e., Kaggle credit dataset and MNIST dataset). In addition, we evaluate G-PATE on the Fashion-MNIST dataset consisting of real-world images of clothes.

Table 1: **Performance Comparison on Kaggle Credit Dataset.** The table presents AUROC of the classifier trained on synthetic data and tested on real data. The evaluation results for PATE-GAN and DP-GAN are recorded in Yoon et al. (2019). We evaluate G-PATE under the same experimental setup. PATE-GAN, DP-GAN, and G-PATE all satisfy $(1, 10^{-5})$-differential privacy. The best results out of different models are bolded.

|  | GAN | PATE-GAN | DP-GAN | G-PATE |
|---|---|---|---|---|
| **Logistic Regression** | 0.9430 | 0.8728 | 0.8720 | **0.9251** |
| **AdaBoost (Freund et al., 1996)** | 0.9416 | 0.8959 | 0.8809 | **0.8981** |
| **Bagging (Breiman, 1996)** | 0.9379 | 0.8877 | 0.8657 | **0.8964** |
| **Multi-layer Perceptron** | 0.9444 | 0.8925 | 0.8787 | **0.9093** |
| **Average** | 0.9417 | 0.8872 | 0.8743 | **0.9072** |

## 6.1 EXPERIMENTAL SETUP

To compare with PATE-GAN, we use the same Kaggle credit card fraud detection dataset (Dal Pozzolo et al., 2015) (Kaggle Credit) as in Yoon et al. (2019) [1]. The dataset contains 284,807 samples representing transactions made by European cardholders' credit cards in September 2013, and 492 (0.2%) of these samples are fraudulent transactions. Each sample consists of 29 continuous features which are the results of a PCA transformation from the original features.

To demonstrate the superiority of G-PATE to PATE-GAN on high dimensional image datasets, we train G-PATE on the MNIST and Fashion-MNIST datasets (Xiao et al., 2017). Each of them consists of 60,000 training examples and 10,000 testing examples. Each example is a $28 \times 28$ grayscale image, associated with a label from 10 classes. The examples in the MNIST dataset are images of handwritten digits between 0 and 9, and the examples in the Fashion-MNIST dataset are real-world images of clothes taken from the Zalando articles. Fashion-MNIST is proposed as a replacement for the MNIST dataset because it better represents modern computer vision tasks.

For the Kaggle Credit dataset, both the generator and discriminator networks of G-PATE are fully connected neural network with the same architecture as PATE-GAN (Yoon et al., 2019). We use random projection with 5 projected dimensions during gradient aggregation. We use the DCGAN (Radford et al., 2015) structure on both MNIST and Fashion-MNIST. We use random projection with 10 projected dimensions during gradient aggregation. More details are provided in Appendix A.5.

## 6.2 COMPARISON WITH DP-GAN AND PATE-GAN

**Kaggle Credit.** The Kaggle Credit dataset is highly unbalanced. In PATE-GAN, the ratio between positive and negative classes in the sensitive training set is assumed to be public information. In contrast, the G-PATE method does not rely on any public information about the sensitive training dataset. It calculates the ratio between positive and negative classes using Laplacian mechanism (Dwork et al., 2014) with $\varepsilon = 0.01$. We then train a $(0.99, 10^{-5})$-differentially private data generator and sample the synthetic records according to the noisy class ratios. By the composition theorem of differential privacy (Dwork et al., 2014), the data generation mechanism is $(1, 10^{-5})$-differentially private.

To compare with PATE-GAN, we select 4 commonly used classifiers evaluated in Yoon et al. (2019). The performance of a generator is measured by the AUROC of the 4 classifiers trained on the corresponding synthetic data. We evaluate G-PATE under the same experimental setups as PATE-GAN for $\varepsilon = 1$.[2] The results for PATE-GAN and DP-GAN are from Yoon et al. (2019), and we get a higher baseline performance from GAN compared to the baseline performance reported in Yoon et al. (2019).

Table 6 presents the comparative analysis between G-PATE and PATE-GAN on Kaggle Credit dataset. G-PATE outperforms both PATE-GAN and DP-GAN and is close to the original GAN which has no privacy protection. The good performance of G-PATE is partly due to the relatively low dimensionality of the Kaggle Credit dataset and the abundance of training examples. More experimental results on Kaggle Credit dataset are presented in Appendix A.1.

---

[1]PATE-GAN does not open source, so we directly compare with the results they reported.

[2]We reproduce the experimental setups of PATE-GAN to the best of our knowledge according to the paper.

Table 2: **Performance Comparison on Image Datasets.** We compare G-PATE with DP-GAN and GAN on MNIST and Fashion-MNIST datasets. The table presents the 10-class classification accuracy of a model trained on synthetic data and tested on real data. DP-GAN and G-PATE are both evaluated under two private settings: $\varepsilon = 1, \delta = 10^{-5}$ and $\varepsilon = 10, \delta = 10^{-5}$.

| Dataset | GAN | DP-GAN | G-PATE |
|---------|-----|--------|--------|
| **MNIST** | 0.9653 ($\varepsilon = \infty$) | 0.4036 ($\varepsilon = 1$) | **0.5631** ($\varepsilon = 1$) |
| | | 0.8011 ($\varepsilon = 10$) | **0.8092** ($\varepsilon = 10$) |
| **Fashion-MNIST** | 0.8032 ($\varepsilon = \infty$) | 0.1053 ($\varepsilon = 1$) | **0.5174** ($\varepsilon = 1$) |
| | | 0.6098 ($\varepsilon = 10$) | **0.6934** ($\varepsilon = 10$) |

**MNIST and Fashion-MNIST.** To understand G-PATE's performance on image datasets, we perform comparative analysis between G-PATE and DP-GAN on the MNIST and Fashion-MNIST datasets[3]. We evaluate the generator by the 10-class classification accuracy of models trained on synthetic data and tested on real data (Table 2). The analysis is performed under two performance settings: $\varepsilon = 1, \delta = 10^{-5}$ and $\varepsilon = 10, \delta = 10^{-5}$. G-PATE outperforms DP-GAN under both settings, and there is a more significant improvement for the setting with stronger privacy guarantee (i.e., $\varepsilon = 1$). Specifically, we observe that DP-GAN fails to converge on the Fashion-MNIST dataset with $\varepsilon = 1$. The synthetic records generated by DP-GAN under this setting are close to random noise while the model trained on G-PATE generated data retains an accuracy of 51.74%.

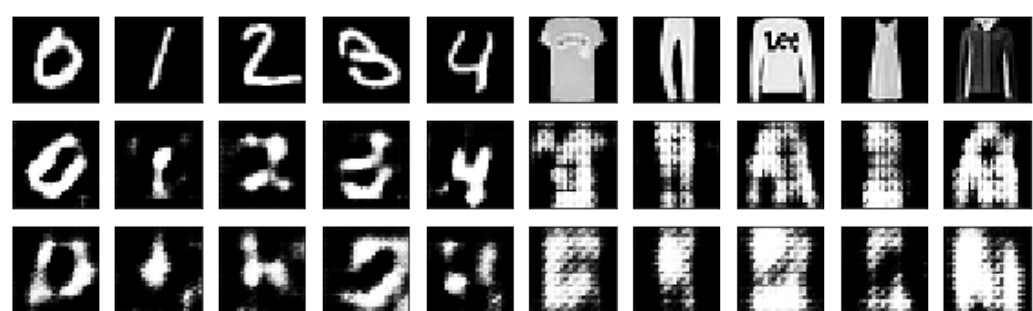

Figure 2: **Visualization of generated instances by G-PATE.** Row 1 (real image), row 2 ($\varepsilon = 10, \delta = 10^{-5}$) and row 3 ($\varepsilon = 1, \delta = 10^{-5}$) each presents one image from each class (the left 5 columns are MNIST images, and the right 5 columns are Fashion-MNIST images). When $\varepsilon = 1$, G-PATE does not generate high-quality images. However, it preserves partial features in the training images, so the synthetic images are useful to preserve data utility which can be seen from our quantitative results.

**Analysis on the Number of Teachers and the Projection Dimensions.** We perform comprehensive ablation studies on the number of teachers and the the projection dimensions to gain better understanding about G-PATE. As shown in Table 3, G-PATE benefits from having more teacher discriminators. Under the same privacy guarantee, the number of noisy votes ($\sigma_1$ and $\sigma_2$) remains the same, so the output of the noisy voting algorithm is more likely to be correct, and the model would get better performance. However, this benefit diminishes as the training set for each teacher model gets smaller with the increasing number of teachers, and 4000 teachers have already achieved satisfiable results. Table 3 also demonstrates the effectiveness of the random projection method, which improves the classification accuracy by around 0.45.

---

[3]We are unable perform comparative study with PATE-GAN on image datasets because PATE-GAN does not report any results on images.

Table 3: **Analysis on the Number of Teachers and the Projection Dimensions.** We performed comprehensive studies on the number of teachers and the the projection dimensions on MNIST with $\varepsilon = 1$ and $\delta = 10^{-5}$. The model has the best performance with 4000 teacher models and projection dimension equals to 10.

| | Projection Dimensions | | | | # of Teachers | | |
|---|---|---|---|---|---|---|---|
| | 5 | 10 | 20 | No Projection | 2000 | 3000 | **4000** |
| **MNIST** | 0.4638 | **0.5631** | 0.5604 | 0.1141 | 0.4240 | 0.5218 | **0.5631** |
| **Fashion** | 0.5129 | **0.5174** | 0.5172 | 0.1268 | 0.3997 | 0.4874 | **0.5174** |

## 7 CONCLUSION

This paper proposes G-PATE, a novel approach for training a differentially private data generator by ensuring privacy property on the information flow from the discriminator to generator in GAN. G-PATE is enabled by a differentially private gradient aggregation mechanism combined with random projection. It significantly outperforms prior work on both image and non-image datasets. Moreover, G-PATE retains reasonable utility on complex image dataset for which DP-GAN can hardly converge.

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

## A   APPENDIX

### A.1   ADDITIONAL EVALUATION RESULTS ON KAGGLE CREDIT DATASET

In addition to AUROC, we also evaluate the AUPRC of the classification models trained on the synthetic data produced by different generative models. Table 4 presents the results. G-PATE has the best performance among all the differentially private generative models.

|  | GAN | PATE-GAN | DP-GAN | G-PATE |
|---|---|---|---|---|
| **Logistic Regression (LR)** | 0.4069 | 0.3907 | 0.3923 | **0.4476** |
| **AdaBoost** | 0.4530 | 0.4366 | 0.4234 | **0.4481** |
| **Bagging** | 0.3303 | 0.3221 | 0.3073 | **0.3503** |
| **Multi-Layer Perceptron (MLP)** | 0.4790 | 0.4693 | 0.4600 | **0.5109** |
| **Average** | 0.4173 | 0.4046 | 0.3958 | **0.4392** |

Table 4: **AUPRC on Kaggle Credit Dataset.** The table presents AUPRC of classification models trained on synthetic data and tested on real data. PATE-GAN, DP-GAN, and G-PATE all satisfy $(1, 10^{-5})$-differential privacy. The best results among different DP generative models are bolded.

To understand the upper-bound of the classification models' performance. We train the same classification models on real data and test it on real data. The results are presented in Table 5.

|  | LR | AdaBoost | Bagging | MLP |
|---|---|---|---|---|
| **AUROC** | 0.9330 | 0.9802 | 0.9699 | 0.9754 |
| **AUPRC** | 0.6184 | 0.7103 | 0.6707 | 0.8223 |

Table 5: **Performance of Classification Models Trained on Real Data.** The table presents AUROC and AUPRC of classification models trained and tested on real data. These results are the upper-bounds for evaluation results on Kaggle Credit dataset.

### A.2   SYNTHETIC IMAGES GENERATED BY G-PATE

Figure 3 presents the synthetic images generated by G-PATE on MNIST and Fashion-MNIST. Images in the same column share the same class label. Row 1 contain real images in the training dataset; row 2 contain images generated by G-PATE when $\epsilon = 10, \delta = 10^{-5}$; and row 3 contain images generated by G-PATE when $\epsilon = 1, \delta = 10^{-5}$.

### A.3   PERFORMANCE ANALYSIS ON NONPRIVATE GPATE

To understand how the GPATE training framework influence the performance of a GAN, we train a nonprivate GPATE with 10 teacher models. As shown in Table 6, the GPATE structure has a comparable performance to the vanilla GAN.

Table 6: **Performance Comparison between GAN and nonprivate GPATE on Kaggle Credit Dataset.**

|  | GAN | Nonprivate GPATE |
|---|---|---|
| **Logistic Regression** | 0.9430 | 0.9455 |
| **AdaBoost (Freund et al., 1996)** | 0.9416 | 0.9165 |
| **Bagging (Breiman, 1996)** | 0.9379 | 0.9456 |
| **Multi-layer Perceptron** | 0.9444 | 0.9219 |
| **Average** | 0.9417 | 0.9324 |

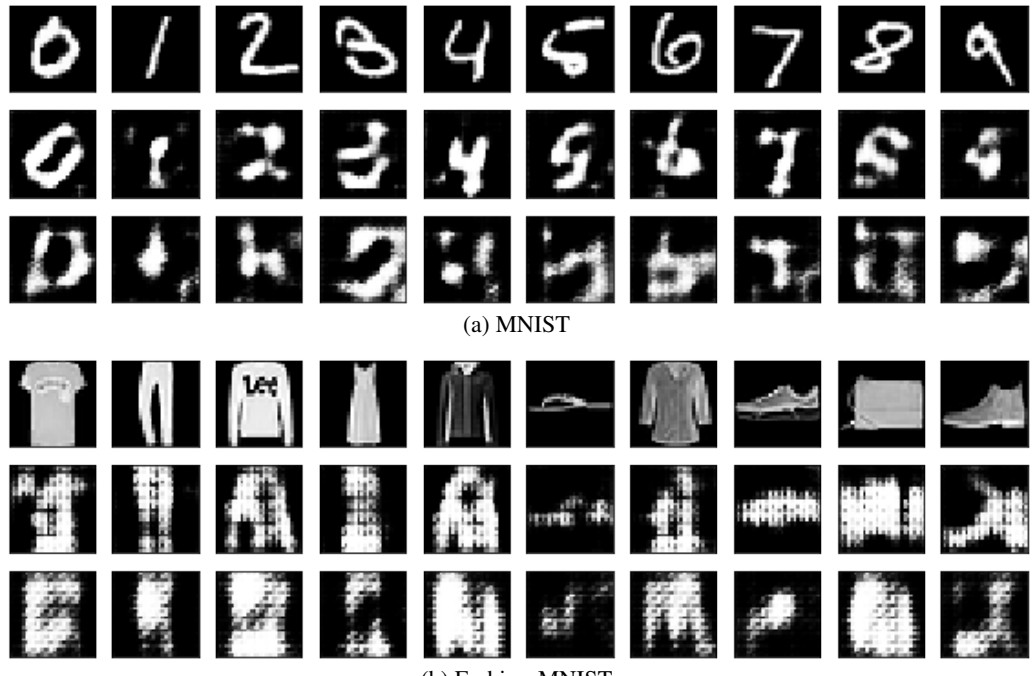

(a) MNIST

(b) Fashion-MNIST

Figure 3: **Visualization of generated instances by G-PATE.** Row 1 (real image), row 2 ($\varepsilon = 10, \delta = 10^{-5}$) and row 3 ($\varepsilon = 1, \delta = 10^{-5}$) each presents one image from each class.

## A.4 PRIVACY BUDGET OF CONFIDENT-GNMAX

The `Confident-GNMax` aggregator was proposed by Papernot et al. (2018) to support differentially private aggregation of the votes from multiple teacher models. For the completeness of this paper, in this section, we include the algorithm for the `Confident-GNMax` aggregator and its data-dependent RDP guarantee.

### A.4.1 THE CONFIDENT-GNMAX AGGREGATOR

---

**Algorithm 3 Confident-GNMax Aggregator.** The private aggregator used in the scalable PATE framework Papernot et al. (2018).

---

**Require:** input $x$, threshold $T$, noise parameters $\sigma_1$ and $\sigma_2$
1: **if** $\max_i \{n_j(x)\} + \mathcal{N}(0, \sigma_1^2) \geq T$ **then**
2:     **return** $\arg\max\{n_j(x) + \mathcal{N}(0, \sigma_2^2)\}$
3: **else**
4:     **return** $\perp$
5: **end if**

---

Algorithm 3 presents the `Confident-GNMax` aggregator proposed by Papernot et al. (2018). The algorithm contains two steps. First, it computes the noisy maximum vote

$$M_1 = \max_i \{n_j(x)\} + \mathcal{N}(0, \sigma_1^2).$$

Then, if the noisy maximum vote is greater than a given threshold, it uses the GNMax mechanism to select the output with most votes:

$$M_2 = \arg\max\{n_j(x) + \mathcal{N}(0, \sigma_2^2)\}.$$

Since each teacher model may cause the maximum number of vote to change at most by 1, $M_1$ is equivalent to a Gaussian mechanism with sensitivity 1. Therefore, following Theorem 3, $M_1$ with Gaussian noise of variance $\sigma_1^2$ guarantees $(\lambda, \lambda/2\sigma_1^2)$-RDP for all $\lambda > 1$.

$M_2$ could be decomposed into post-processing a noisy histogram with Gaussian noise added to each dimension. Since each teacher model may increase the count in one bin and decrease the count in another, the mechanism has a sensitivity of 2. Therefore, $M_2$ with Gaussian noise of variance $\sigma_2^2$ guarantees $(\lambda, \lambda/\sigma_2^2)$-RDP (Papernot et al., 2018).

The data-dependent privacy guarantee for the GNMax mechanism $M_2$ has been analyzed by Papernot et al. (2018):

**Theorem 7.** *Let $M$ be a randomized algorithm with $(\mu_1, \varepsilon_1)-RDP$ and $(\mu_2, \varepsilon_2)-RDP$ guarantees and suppose that there exists a likely outcome $i^*$ given a dataset $D$ and a bound $\tilde{q} \leq 1$ such that $\tilde{q} \geq \Pr\left[\mathcal{M}(D) \neq i^*\right]$. Additionally, suppose that $\lambda \leq \mu_1$ and $\tilde{q} \leq e^{(\mu_2-1)\varepsilon_2} / \left( \frac{\mu_1}{\mu_1 - 1} \cdot \frac{\mu_2}{\mu_2 - 1} \right)^{\mu_2}$. Then, for any neighboring dataset $D'$ of $D$, we have:*

$$D_\lambda\left(\mathcal{M}(D)\|\mathcal{M}\left(D'\right)\right) \leq \frac{1}{\lambda - 1} \log\left( (1 - \tilde{q}) \cdot \boldsymbol{A}\left(\tilde{q}, \mu_2, \varepsilon_2\right)^{\lambda-1} + \tilde{q} \cdot \boldsymbol{B}\left(\tilde{q}, \mu_1, \varepsilon_1\right)^{\lambda-1} \right),$$

*where $\boldsymbol{A}\left(\tilde{q}, \mu_2, \varepsilon_2\right) \triangleq (1 - \tilde{q})/\left(1 - (\tilde{q}e^{\varepsilon_2})^{\frac{\mu_2-1}{\mu_2}}\right)$ and $\boldsymbol{B}\left(\tilde{q}, \mu_1, \varepsilon_1\right) \triangleq e^{\varepsilon_1}/\tilde{q}^{\frac{1}{\mu_1-1}}$.*

The parameters $\mu_1$ and $\mu_2$ are optimized to get a data-dependent RDP guarantee for any order $\lambda$. By applying Theorem 7 on $M_2$, we obtain the data-dependent RDP budget for $M_2$.

For any $\lambda > 1$, suppose $\varepsilon_1$ is the RDP budget for $M_1$ and $\varepsilon_2$ is the data-dependent RDP budget for $M_2$. Then, the RDP budget for the `Confident-GNMax` algorithm could be calculated as follows:

$$\varepsilon = \begin{cases} \varepsilon_1 & \text{if output is } \perp, \\ \varepsilon_1 + \varepsilon_2 & \text{otherwise.} \end{cases}$$

### A.5 MODEL STRUCTURES AND HYPERPARMETERS

**G-PATE.** For MNIST and Fashion-MNIST, the student generator consists of a fully connected layer with 1024 units and a deconvolutional layer with 64 kernels of size $5 \times 5$ (strides $2 \times 2$). Each teacher discriminator has a convolutional layer with 32 kernels of size $5 \times 5$ (strides $2 \times 2$) and a fully connected layer with 256 units. All layers are concated with the one-hot encoded class label. We apply batch normalization and `Leaky ReLU` on all layers. When $\varepsilon = 10$, we train 2000 teacher discriminators with batch size of 30 and set $\sigma_1 = 600, \sigma_2 = 100$. When $\varepsilon = 1$, we train 4000 teacher discriminators with batch size of 15 and set $\sigma_1 = 3000, \sigma_2 = 1000$. For Kaggle Credit dataset, we train 2100 teacher discriminators with batch size of 32 and set $\sigma_1 = 1500, \sigma_2 = 600$. For all three datasets, we use Adam optimizer Kingma & Ba (2014) with learning rate of $10^{-3}$ to train the models and clip the adversarial perturbations between $\pm 10^{-4}$. The consensus threshold $T$ is set to 0.5.

**GAN.** The structure of GAN is the same as the structure of G-PATE with a single teacher discriminator. The hyper-parameters are also the same as G-PATE.

**DP-GAN.** We use DP-GAN method mentioned in Xie et al. (2018) on both MNIST and FashionMNIST tasks. For the generator, we use FC Net structure with [128, 256, 512, 784] neurons in each layer, and the discriminator contains [784, 64, 64, 1] neurons in each layer. In each training epoch, the discriminator trains 5 steps and the generator trains 1 step. For both networks, $0.5 \times \text{ReLU}(\cdot)$ activation layers are used. Our batch size is 64 for each sampling, and sampling rate $q$ equals to $\frac{64}{6 \times 10^4}$. We bound the discriminator's parameter weights to $[-0.1, 0.1]$ and kept feature's value between $[-0.5, 0.5]$ during the forward process. In order to generate specific digit data, we concat one-hot vector, which represents digits categories, into each layer in both the disciminator and the generator.

**Classification Models for MNIST and Fashion-MNIST.** For each synthetic dataset, we trian a CNN for the classification task. The model has two convolutional layers with 32 and 64 kernels respectively. We use `ReLU` as the activation function and applies dropout on all layers.

**Classification Models for Kaggle Credit.** We implement 4 predictive models in Yoon et al. (2019) using sklearn: Logistic Regression (`LogisticRegression`), Adaptive Boosting (`AdaBoostClassifier`), Bootstrap Aggregating (`BaggingClassifier`) and Multi-layer

Perceptron (`MLPClassifier`). We use *L1* penalty, `Liblinear` solver and *(350:1)* class weight in Logistic Regression. We use logistic regression as classifier in Adaptive Boosting and Bootstrap Aggregating, setting *L2* penalty, number of models as 200 and 100. For Multi-layer Perceptron, we use `tanh` as the activation of 3 layers with 18 nodes and Adam as the optimizer.

