# OpenReview forum: "Scalable Differentially Private Data Generation via Private  Aggregation  of  Teacher Ensembles"
_ICLR.cc/2020/Conference — Reject_

### Official Review · AnonReviewer2 · 2019-10-24
**Official Blind Review #2**

**Rating:** 3

**Review:**

This paper studies the problem of differentially private data generator. Inspired by the general GAN framework and the PATE mechanism, the authors propose a new differentially private training algorithm for data generator. The problem of training data generator with privacy guarantee considered in this paper is very interesting, and the proposed algorithm looks novel. However, there are lots of unclear statements in the current paper, and I cannot tell whether the proposed algorithm is indeed better than previous methods. Following are my major concerns:
1.It is unclear what are the loss functions used in equation (1) and (2). Please define k when introducing equation (2).
2.The training framework introduced in section 3.1 is different from the traditional GAN framework, and thus my concern is that whether this framework will give us good generated samples. Because the performance of GAN has been proved in both theory and practice. The authors should at least empirically show the performance of the proposed framework in the nonprivate setting.
3.There is no introduction of the $(\epsilon,\delta)$-differential privacy before introducing the Definition 1.
4.There is no definition of Renyi differential privacy, so the statement of Theorem 2 is unclear. In addition, what is data-dependent Renyi differential privacy?
5.The privacy guarantee of Algorithm 2 is not very clear. Because there are lots of parameters in Algorithm 1 and 2 which may affect the privacy guarantee, and Theorem 3 does not state such requirements. For example, how to choose $\sigma_1,\sigma_2$? In Theorem 7, there are some constraints on different parameters, will them be satisfied by your algorithm?
6.How will the number of teacher models affect the privacy guarantee?
7.Why you choose random projection matrix with variance $1/k$, and what is the projection dimension for different algorithms?
8.In Table 1, the results of non private GAN are different from the results of non private GAN reported in PATE-GAN paper. Since the baseline results are much better in the current than the results reported in the PATE-GAN paper, it seems to me that the improvements of the proposed method comes from the stronger baseline.

Other comments:
1.$\lambda>1$ in Theorem 3.
2.Algorithm 2 should be moved to main context.
3.The last sentence in section 3.2 is not convincing.
4.Typo “differnet” in the caption of Table 1.

**Experience Assessment:**

I have published one or two papers in this area.

**Review Assessment: Checking Correctness Of Derivations And Theory:**

I assessed the sensibility of the derivations and theory.

**Review Assessment: Checking Correctness Of Experiments:**

I carefully checked the experiments.

**Review Assessment: Thoroughness In Paper Reading:**

I read the paper thoroughly.

---

> ### Author Response · Authors · 2019-11-14
> **Response to Review Questions**
>
>
> > 1.It is unclear what are the loss functions used in equation (1) and (2).
> Thanks for pointing it out. The loss function could be any loss function that is applicable for the discriminator, and k is the number of synthetic record generated per training iteration. We will update the paper to include the explanations before equation (1) and (2).
>
> > 2. Performance of GPATE under nonprivate setting.
> Thanks for the interesting suggestion. We conducted additional experiments (Table 6 in Section A.3 in the revision). As shown in the results, the performance of nonprivate GPATE is comparable to vanilla GAN, which indicates that the structure of GPATE will not have a strong impact on the data generation process.
>
> > 3 & 4. There is no introduction to differential privacy and Renyi differential privacy.
> Thanks for the suggestion! We included definitions for differential privacy and Renyi differential privacy in Appendix A.5 due to the space limit. We will move them to the background section in the revision.
> In particular, Data-dependent Renyi differential privacy (RDP) is RDP with privacy budget depending on the data.
>
> > 5. There are lots of parameters in Algorithm 1 and 2 which may affect the privacy guarantee, and Theorem 3 does not state such requirements. In Theorem 7, there are some constraints on different parameters, will them be satisfied by your algorithm?
> Thanks for the interesting question. $\sigma_1$, $\sigma_2$, and threshold $T$ are parameters for the Confident-GNMax algorithm proposed by Papernot et al. (2018). Their influence on the privacy budget is included in Section A.5 (A.4 in the revision). The constraints in Theorem 7 are not used to select algorithm parameters. Instead, they are used to calculate the data dependent privacy budget for the Confident-GNMax algorithm. Under the constraints, the parameters $\mu_1$ and $\mu_2$ are optimized to get a data-dependent RDP guarantee for any order $\lambda$. We omitted this process because this guarantee is addressed in the similar way in prior work by Papernot et al. 2018. We have added the corresponding analysis in the Appendix.
> Theorem 2 (Theorem 4 in the revision) analyzes how the training iteration n and projection dimension k influences the privacy guarantee. The remaining parameters do not influence the privacy guarantee. We have added this analysis under the Theorem, and thanks for the suggestions!
>
> > 6. How will the number of teacher models affect the privacy guarantee?
> We have analyzed the influence of the number of teacher models in Section 5.2 (Section 6.2 in the revision) in Table 3. It shows that, under the same privacy guarantee, increasing the number of teacher models would improve model performance. Equivalently, under the same level of model performance, increasing the number of teacher models could improve privacy guarantee.
> Theoretically, the number of teacher models do not directly influence the privacy guarantee; while with more teacher models there are more votes in each bin. We provide the intuition based on the following two scenarios: (1) Under the same privacy guarantee, the number of noisy votes ($\sigma_1$ and $\sigma_2$) remains the same. Therefore, the output of the noisy voting algorithm is more likely to be correct, and the model would get better performance. (2) Under the same level of model performance, the algorithm could tolerate higher number of noisy votes ($\sigma_1$ and $\sigma_2$), so the privacy guarantee is improved. We will add such analysis in the revision.
>
> > 7.Why you choose random projection matrix with variance 1/k, and what is the projection dimension for different algorithms?
> Random projection matrix with variance 1/k is a standard trick in differential privacy (Xu et al. 2017).
> Based on the dimensionality of the original data, we set the projection dimension to be 10 for both MNIST and Fashion-MNIST, and 5 for Kaggle Credit. (Please refer to Section 5.1 in the submitted version / Section 6.1 in the revision).
> We also conducted additional ablation study on the projection dimensions on Fashion-MNIST (Table 3 in the revision).
>
> > 8. Since the baseline results are much better in the current than the results reported in the PATE-GAN paper, it seems to me that the improvements of the proposed method comes from the stronger baseline.
> We train GPATE and vanilla GAN separately, so GPATE does not benefit from the better performance of vanilla GAN. Additionally, we use the same architecture as the PATE-GAN paper, and compare the baselines under the same experimental settings. We have shared our code and we are happy to have the community to run our pipeline and evaluate the performance.
>
> > Other comments
> We have corrected the typos in the revision and thank you for pointing them out.
>
> Reference:
> Xu, Chugui, et al. "DPPro: Differentially private high-dimensional data release via random projection." IEEE Transactions on Information Forensics and Security 12.12 (2017): 3081-3093.

---

### Official Review · AnonReviewer1 · 2019-10-26
**Official Blind Review #1**

**Rating:** 1

**Review:**

In this submission, the authors propose a method to generate synthetic datasets with privacy guarantee while preserving high data utility. However, the following concerns are important to clarify for the authors:

1) Compared to existing work, the proposed method can preserve high data utility due to the fact that the proposed method doesn't ensure differential privacy for the discriminator. Could the authors provide more details about this key observation and motivation? How about discriminator is not trustable? Can we simply assume that it is safe for discriminator to access sensitive data? How about the discriminator is attacked? I agree for some applications, we can have this assumption about discriminator, but it is very important to better understand the limitation and risk of this assumption. The real-world applications are not simply defined by us.

2) The authors claim that the proposed method is scalable? Could the author confirm this either theoretically or experimentally?

3) The experiments are conducted on a single dataset, simple MNIST dataset. It would be convincing to report experiments on more datasets such as CIFAR-10 and others.

4) For experiment comparison and analysis, the authors adopt quite large epsilon, i.e., \epsilon = 1 and \epsilon = 10. Several existing work adopt \epsilon = 0.2. Can the authors report experiment comparison with such meaningful epsilon?

**Experience Assessment:**

I have published one or two papers in this area.

**Review Assessment: Checking Correctness Of Derivations And Theory:**

I assessed the sensibility of the derivations and theory.

**Review Assessment: Checking Correctness Of Experiments:**

I carefully checked the experiments.

**Review Assessment: Thoroughness In Paper Reading:**

I read the paper thoroughly.

---

> ### Author Response · Authors · 2019-11-14
> **Response to Review Questions**
>
>
> > (1) What if the discriminator is attacked?
> Thanks for the questions. Just as any GAN based generative model, in GPATE the discriminator is NOT shared to public (please refer to Section 3 paragraph 2 in the submitted version / Section 4 in the revision). Therefore, launching any attack on the discriminator is not possible because the attacker would not have access to the discriminator.
> In addition, to the best of our knowledge, none of the applications of GAN needs the discriminator in the data generation phase. Therefore, not sharing the discriminator would not influence the application of GPATE.
>
> > (2) Could the authors confirm the scalability of the approach experimentally?
> GPATE is more scalable compared to its prior work DP-GAN and PATE-GAN. This has been demonstrated by its performance on two image datasets: MNIST and Fashion-MNIST, while the state of art baselines can only be performed on low dimension non-image dataset (Xie et al. 2018, Yoon et al. 2019).
>
> > (3) The experiments are conducted on a single dataset.
> We have conducted experiments on three datasets in the paper: Kaggle Credit, MNIST, and Fashion-MNIST, which is shown in Section 5 (Section 6 in the revision). Table 1 and Table 2 have bolded the names of the datasets. We will make this more clear in the revision.
>
> > (4) The authors adopt quite large epsilon.
> $\epsilon = 1$ has been accepted as a standard privacy protection criterion in many prior work on differential privacy (Papernot et al. 2017, Papernot et al. 2018, Yoon et al. 2019).
>
> References:
>
> Nicolas Papernot, Martín Abadi, Ulfar Erlingsson, Ian Goodfellow, and Kunal Talwar. Semi- supervised knowledge transfer for deep learning from private training data. In International Conference on Learning Representations, 2017. URL https://openreview.net/forum?id=HkwoSDPgg.
>
> Liyang Xie, Kaixiang Lin, Shu Wang, Fei Wang, and Jiayu Zhou. Differentially private generative adversarial network. arXiv preprint arXiv:1802.06739, 2018.
>
> Jinsung Yoon, James Jordon, and Mihaela van der Schaar. PATE-GAN: Generating synthetic data with differential privacy guarantees. In International Conference on Learning Representations, 2019. URL https://openreview.net/forum?id=S1zk9iRqF7.

---

### Official Review · AnonReviewer3 · 2019-10-27
**Official Blind Review #3**

**Rating:** 8

**Review:**

This paper presents a framework for Differentially private data generation that enables more accurate training of downstream learners without compromising on the privacy guarantees. The key insight is to introduce an ensemble of teacher discriminators in the GAN formulation instead of a single discriminator. The teachers are trained on separate subsets of the private data set. A gradient aggregator is also introduced for transmitting loss signal to the student without losing privacy by using adversarial perturbations and random projections. The authors provide theoretic guarantees of Renyi differential privacy and experiments on Kaggle Credit default dataset and MNIST.

I have very little knowledge of this field, but the main idea idea seemed quite novel and insightful.  I have not verified the theoretical results closely, I only skimmed the derivations, but what I can understand, it seems reasonable. The experimental results seemed quite good, with improvements across the board. I would have liked to see some metric that  was not just performance of the downstream classifier, is there an intrinsic measure of the generator's performance that makes sense to report?

The presentation was quite reasonable and polished, with very few typos.  My only gripe is that they seemed to spend a little too much space in different sections re-iterating their key contributions, but not enough defining or citing sources for key definitions such as Renyi differential privacy, which would be helpful to a non-expert reader.



**Experience Assessment:**

I do not know much about this area.

**Review Assessment: Checking Correctness Of Derivations And Theory:**

I did not assess the derivations or theory.

**Review Assessment: Checking Correctness Of Experiments:**

I assessed the sensibility of the experiments.

**Review Assessment: Thoroughness In Paper Reading:**

I made a quick assessment of this paper.

---

> ### Author Response · Authors · 2019-11-14
> **Response to Review Questions**
>
>
> > Is there an intrinsic measure of the generator's performance that makes sense to report?
> Thanks for the interesting question.
> Given that our main goal of GPATE is to generate differentially private data, we followed the state of the art private generative model evaluation setup (Xie et al. 2018, Yoon et al. 2018): 1. We theoretically analyze the privacy guarantees; 2. We empirically evaluate the utility of the generated data by evaluating the trained models (e.g. classifiers) on real data.
> In addition, evaluating the quality of the generated instances (e.g. images) is challenging in general, and the community is still trying to propose better metric for evaluation.
>
> > Not enough defining or citing source for key definitions such as Renyi differential privacy.
> Thanks for pointing this out.
> We included definitions for differential privacy and Renyi differential privacy in Appendix A.5 due to space limit. We have moved them to the background section in the revision.
>
> References:
> Liyang Xie, Kaixiang Lin, Shu Wang, Fei Wang, and Jiayu Zhou. Differentially private generative adversarial network. arXiv preprint arXiv:1802.06739, 2018.
>
> Jinsung Yoon, James Jordon, and Mihaela van der Schaar. PATE-GAN: Generating synthetic data with differential privacy guarantees. In International Conference on Learning Representations, 2019. URL https://openreview.net/forum?id=S1zk9iRqF7.

---

### Author Response · Authors · 2019-11-14
**Revisions to Address Requests**

We thank the reviewers for their valuable feedback. We have made the following changes in our revision:
- We moved the definitions of differential privacy and Renyi differential privacy from the Appendix to Section 3.
- We moved algorithm 2 from the Appendix to Section 4.
- We added additional experiment results in Section 6.
- We added the performance of nonprivate GPATE in the Appendix (Table 6 in Section A.3).

---

### Decision · Program_Chairs · 2019-12-19

**Decision:**

Reject

**Comment:**

This paper addresses the problem of differential private data generator. The paper presents a novel approach called G_PATE which builds on the existing PATE framework. The main contribution is in using a student generator with an ensemble of teacher discriminators and in proposing a new private gradient aggregation mechanism which ensures differential privacy in the information flow from discriminator to generator.

Although the idea is interesting, there are significant concerns raised by the reviewers about the experiments and analysis done in the paper which seem to be valid and have not been addressed yet in the final revision. I believe upon making significant changes to the paper, this could be a good contribution. Thus, as of now, I am recommending a Rejection.